# Minimum Message Length in Hybrid ARMA and LSTM Model Forecasting

**DOI:** 10.3390/e23121601

**Published:** 2021-11-29

**Authors:** Zheng Fang, David L. Dowe, Shelton Peiris, Dedi Rosadi

**Affiliations:** 1Department of Data Science and Artificial Intelligence, Monash University, Clayton, VIC 3800, Australia; zfan51@student.monash.edu; 2School of Mathematics and Statistics, University of Sydney, Camperdown, NSW 2006, Australia; shelton.peiris@sydney.edu.au; 3Department of Statistics, Gadjah Mada University, Sleman, Yogyakarta 55500, Indonesia; dedirosadi@gadjahmada.edu

**Keywords:** long short-term memory, minimum message length, time series, neural network, deep learning, Bayesian statistics, probabilistic modeling

## Abstract

Modeling and analysis of time series are important in applications including economics, engineering, environmental science and social science. Selecting the best time series model with accurate parameters in forecasting is a challenging objective for scientists and academic researchers. Hybrid models combining neural networks and traditional Autoregressive Moving Average (ARMA) models are being used to improve the accuracy of modeling and forecasting time series. Most of the existing time series models are selected by information-theoretic approaches, such as AIC, BIC, and HQ. This paper revisits a model selection technique based on Minimum Message Length (MML) and investigates its use in hybrid time series analysis. MML is a Bayesian information-theoretic approach and has been used in selecting the best ARMA model. We utilize the long short-term memory (LSTM) approach to construct a hybrid ARMA-LSTM model and show that MML performs better than AIC, BIC, and HQ in selecting the model—both in the traditional ARMA models (without LSTM) and with hybrid ARMA-LSTM models. These results held on simulated data and both real-world datasets that we considered.We also develop a simple MML ARIMA model.

## 1. Introduction

Forecasting in time series is a difficult task due to the presence of trends and/or seasonal components. For example, economic time series data are highly impacted by seasonal factors and often show trends with long-run cycles. Such trends and seasonality are difficult to capture by the traditional Autoregressive Moving Average model (ARMA) [1]. The Bayesian Minimum Message Length (MML) principle [2], the Akaike Information Criterion (AIC) [3], Schwarz’s Bayesian information criterion (BIC) [4] and Hannan–Quinn (HQ) [5] are often used in model selection for the ARMA model [6,7,8]. The models selected by MML87 [9] in ARMA time series have lower prediction errors than those from AIC, BIC, and HQ [10]. Schmidt previously showed that MML87 outperforms a variety of other (information-theoretic) approaches in ARMA time series modeling [11] (chapters 5 to 8). In this paper, we extended the traditional ARMA time series model to form the hybrid ARMA-LSTM by combining the neural network of long short-term memory (LSTM) in order to test the performance of MML in model selection. The results suggest that MML outperforms AIC, BIC, and HQ.

The ARIMA is used with integer differencing to achieve stationarity if the time series is not stationary. A time series with seasonal components can be modeled using the family of seasonal ARIMA (or SARIMA) models. On the other hand, this ARIMA family has been generated to include long memory time series using a suitable fractional order differencing in (0, 0.5) to form the family of autoregressive fractionally integrated moving average (ARFIMA) models. Nevertheless, the deep learning LSTM technique might be more suitable to capture the information that is less obvious in the time series, as it allows for a much more general class of models. Time series analysts require a lot of effort to discover the appropriate model in order to identify the dependency in time series data [12]. Historically, the ARMA model was introduced by Box and Jenkins in 1976 [13], and it is popular and widely used in the time series science community and provides accurate forecasts in both in-sample and out-of-sample data when the parameters are correctly estimated [14]. It is a hybrid (or mixture) of autoregressive (AR) and moving average (MA) processes, but the ARMA model can only be used in stationary time series [15].

In parallel, machine learning has seen the development of neural network models in computer science, ultimately influencing statistics. Similar to the families of the ARMA model, deep learning also has several variants, such as a deep neural network (DNN), a convolutional neural network (CNN), and a recurrent neural network (RNN). This report investigates a particular form of the RNN called long short-term memory (LSTM), which is typically used in time series [16]. In recent years, LSTM has been shown to work well in forecasting for data with complex time dependency, such as the stock market and energy consumption prediction [17]. In this paper, we select the best ARMA(*p*,*q*) model and then train the LSTM model for the residuals through the ARMA model. The time-step order used in LSTM is the parameter *q* in ARMA(*p*,*q*) determined by different information-theoretic criteria [18,19].

Our results show that MML compares favorably with the other information-theoretic approaches, including AIC, BIC, and HQ, when conducting ARMA-LSTM. Further, we compare the ARMA-LSTM selected by MML with the ARMA model selected by MML. These results also show that MML outperforms when compared to AIC [20], BIC [20], and HQ [21] in terms of selecting a model with lower prediction error, and this holds whether our modeling is enhanced by LSTM or instead is ARMA unassisted by LSTM. The Bayesian information-theoretic MML principle provides more reliable and highly accurate results in the model selection of the hybrid ARMA-LSTM model than other traditional methods (AIC, BIC, HQ). When doing ARMA without a hybrid with LSTM, MML also performs better than other traditional methods (AIC, BIC, HQ). The best performing method considered is the hybrid MML ARMA-LSTM model. These results hold on simulated data and on the real-world datasets considered.

Section 2 introduces the Box and Jenkins theory for the ARIMA model and discusses its limitations. Section 3 introduces the information-theoretic Minimum Message Length criterion in model selection, and Section 4 introduces the deep learning model LSTM. Section 5 provides the algorithm of the hybrid ARMA-LSTM model, and Section 6 provides the experimental results with a comparison.

## 2. ARIMA Modeling

This section reviews the theory of Autoregressive Integrated Moving Average (ARIMA) modeling from Box and Jenkins (1970) [13,15]. Let $\left\{Y_{t}\right\}$ be a homogeneous nonstationary time series and suppose that the $d^{th}$ (d=1,2,…) difference of the series is stationary and is given by $X_{t}={\left(1-B\right)}^{d}Y_{t}$, where *B* is the backshift operator. Then a stationary ARMA(*p*,*q*) model can be fitted for $\left\{X_{t}\right\}$, satisfying
(1)$$X_{t}=c+\sum_{i=1}^{p}\phi_{i}X_{t-i}+\epsilon_{t}+\sum_{i=1}^{q}\theta_{i}\epsilon_{t-i},$$
where $\left\{\epsilon_{t}\right\}∼WN\left(0,\sigma^{2}\right).$

Let $\phi \left(B\right)=1-\phi_{1}B-\ldots -\phi_{p}B^{p};\;\theta \left(B\right)=1+\theta_{1}B+\ldots +\theta_{q}B^{q},$ be two polynomials of degree *p* and *q*, respectively, such that the zeros of ϕ(B) and θ(B) are outside the unit circle. Then the ARMA(*p*,*q*) in Equation (Equation 1) can be written in a compact form as
(2)$$\phi \left(B\right)X_{t}=c+\theta \left(B\right)\epsilon_{t}.$$

Now the corresponding ARIMA(*p*,*d*,*q*) model for the original series $\left\{Y_{t}\right\}$ is given by
(3)$$\phi \left(B\right){\left(1-B\right)}^{d}Y_{t}=c+\theta \left(B\right)\epsilon_{t}.$$

It is known that ARIMA is a form of a linear regression model with the lag order of time series data and corresponding residuals. In an application where the ARIMA model fits well for the given data, then the corresponding residuals through the model should form a random scatter plot with a constant mean and a constant variance over the time, see, for example, ref. [13]. If the ARIMA model is not well fitted for the data or an incorrect model has been fitted, then the residuals will not show a random scatter plot and instead indicate autocorrelations within the residuals. This reveals that the information hidden in the data has not been completely captured by the fitted ARIMA model, and we consider refitting an alternative ARIMA model [22].

The above family of ARIMA models are also capable of modeling a wide range of seasonal data using slight modifications. A seasonal extension of Model (3) can be written for a set of time series data with seasonality m. Incorporating both the seasonal and nonseasonal components together with additional polynomials, a new model is
(4)$$\phi \left(B\right)\Phi \left(B^{m}\right){\left(1-B\right)}^{d}{\left(1-B^{m}\right)}^{D}Y_{t}=c+\theta \left(B\right)\Theta \left(B^{m}\right)\epsilon_{t},$$
where $\Phi \left(B^{m}\right)=1-\Phi_{1}B^{m}-\ldots -\Phi_{P}B^{mP},\;\Theta \left(B^{m}\right)=1+\Theta_{1}B^{m}+\ldots +\Theta_{Q}B^{mQ}$, and *D* is the degree of seasonal differencing. For simplicity, this is written as
(5)$$Y_{t}∼SARIMA\left(p,d,q\right){\left(P,D,Q\right)}_{m}$$

Model (4) is known as the Seasonal ARIMA or SARIMA model.

To estimate the parameters of Model (4), it is important to identify the changes of variance in the autocorrelation function (ACF) plot of data. This ACF provides an indication of linear dependencies among the observation of time series, which is related to the order of the model. In addition, the corresponding partial autocorrelation function (PACF) can be used to confirm the approximate order required in the model.

In this study, we use non-seasonal ARIMA modeling because the non-seasonal degree of differencing *d* can be predetermined in practice. We consider the stationary time series data. Assuming the data are generated from a mean zero stationary ARMA(*p*,*q*) process with Gaussian errors, we use the fact that the distribution of data is a multivariate Gaussian distribution with mean μ=0.

Suppose that we have a sample of *N* observations $y=(y_{1},...,y_{N})$ generated through Model (2), with c=0, and let $\beta =(\phi_{1},...,\phi_{p},\theta_{1},...,\theta_{q},\sigma^{2})$ be the vector of all the parameters. Then the corresponding unconditional log-likelihood function, L(yβ), can be written as:(6)$$L\left(y\beta \right)=-\frac{N}{2}\log\left(2\pi \sigma^{2}\right)-\frac{1}{2}\log\Sigma -\frac{1}{2\sigma^{2}}y^{T}\Sigma^{-1}y,$$
where Σ is the determinant of Σ and $\sigma^{2}\Sigma$ is the N×N theoretical autocovariance matrix of y.

Gegenbauer Autoregressive Moving Average (GARMA) models have been used to model a general family of time series with long memory and seasonal components. This family can be used for a wide variety of applications in finance, engineering, and weather forecasting. See, for example, ref. [23] for a comprehensive review and [24] and references therein for estimation methods together with applications.

## 3. Minimum Message Length

The Bayesian information-theoretic Minimum Message Length (MML) principle [2,6,7,9,19,25] is based on coding theory and can be thought of in several equivalent ways. It can be thought of in terms of a transmitter encoding a two-part message and transmitting it to a receiver, where the first part of the message contains information encoding the model and the second part of the message encodes the data given the model. The length of the first part of the message can be thought of as the complexity of the model, and the length of the second part of the message (effectively, the statistical negative log-likelihood) is a measure of goodness of fit to the observed data. For example, with X={A,B,C,D}, possible encodings would be, e.g., A=00,B=01,C=10, and D=11, or instead, e.g., A=1,B=01,C=001, and D=0001, with the length of code represented as I(), e.g., with A=00, I(A)=2. The code length is typically (close to) the negative logarithm of the probability.

MML thus gives a quantitative information-theoretic trade-off between model complexity (length of first part of message) and goodness of fit (length of second part of message) [26]. A smaller MML value (or, equivalently, a shorter message length) indicates the model is less complex and highly fitted to the data [6]. In practice, minimizing the message length can be expressed as:(7)$$\underset{\theta \in \Theta }{\arg\min}\;\left\{I\left(\theta \right)+I\left(y^{N}\theta \right)\right\},$$
where I(θ) is the length of encoding the assertion (or model), and $I(y^{N}\theta )$ is the length of encoding the detail (or data given the model). In MML, there is (Bayesian) prior knowledge (or a prior distribution), π, over the parameter space. Following Wallace and Freeman [9], MML has been shown to work well in time series models, such as autoregressive (AR) and moving average (MA) models [18,27,28]. We can thus estimate the parameters [7,9] by minimizing the message length:(8)$$MessLen\left(y,\beta \right)=-\log\left(\frac{h_{3}\left(\beta \right)f\left(y_{1},...,y_{N}\beta \right)\epsilon^{N}}{\sqrt{F(\beta )}}\right)+\frac{k}{2}\left(1+\log\kappa_{k}\right)-\log h_{1}\left(p\right)-\log h_{2}\left(q\right),$$
where ϵ is measuring the accuracy of data, $h_{3}\left(\beta \right)$ is the Bayesian prior distribution over the parameter set β, we model the parameter set β using uniform prior [0, 1] in the stationarity region $h_{3}\left(\beta \right)=1$, and $h_{1}\left(p\right)=2^{-(1+p)}$ and $h_{2}\left(q\right)=2^{-(1+q)}$ are the priors on the (non-negative integer) parameters *p* and *q*, k=p+q+1 is the number of continuous-valued parameters, $f(y_{1},...,y_{N}\beta )$ is the standard statistical likelihood function, L=−logf, F(β) is the expected Fisher Information matrix (of expected second-order partial derivatives of *L*) and is a function of the parameter set β, F(β) is the expected Fisher information, $\kappa_{k}$ is the lattice constant (which accounts for the expected error in the log-likelihood function from ARMA model (Equation (Equation 6)) due to the quantization of the *k*-dimensional space, which is bounded above by $\frac{1}{12}$ and bounded below by $\frac{1}{2\pi e}$. For example, $\kappa_{1}=\frac{1}{12}$, $\kappa_{2}=\frac{5}{36\sqrt{3}}$, $\kappa_{3}=\frac{19}{192*2^{1/3}}$, and $\kappa_{k}\to \frac{1}{2\pi e}$ as k→∞).

Ignoring the $-\log h_{1}\left(p\right)$, $-\log h_{2}\left(q\right)$, and −Nlog(ϵ) terms, the message length for the ARMA model β can also be represented as:(9)$$I\left(y,\beta \right)=-\log h_{3}\left(\beta \right)+\frac{1}{2}\log F\left(\beta \right)+\frac{k}{2}\log\kappa_{k}+\frac{k}{2}-\log f\left(y\beta \right)$$

MML87 is model invariant and avoids explicitly constructing the quantized parameter space [7,8,9,25]. This is used for model selection and parameter estimation by choosing the model that minimizes the message length.

MML has been used for a variety of problems, including clustering and mixture modeling [29,30] ([19] Section 6.8), clustering of protein dihedral angles [31], decision graphs (as an extension of decision trees, allowing for disjunctions, or “or”) [32] (Section 7.2.4 [19]) and multi-way joins in decision graphs with dynamic attributes [33], causal Bayesian nets (or Bayesian networks, or causal nets) ([19] Section 7.4) and Bayesian nets with decision trees in their (leaf) nodes [34,35], inference of probabilistic finite state automata (or probabilistic finite state machines, PFSAs, PFSMs) ([19] Section 7.1) and hierarchical PFSAs [36], and (given sufficient data and time, and based to whatever degree on the above-mentioned inference of Bayesian nets) automation of database normalization [37], etc.

Part of the reason for the above list is the universality of the MML approach [7] ([19] Chapter 2) (seeking the single best theory) and that of the predictive approach (seeking a Bayesian weighted combination of theories) of Solomonoff [38,39] ([40] Section 3.1). The MML approach of Wallace and the algorithmic probability approach of Solomonoff both have many desirable properties, but they can be slow in practice, whereas deep learning often runs relatively quickly. This motivates us to combine these approaches, as we do using the deep learning approach of long short-term memory (LSTM). This gives us something of a combination of the simplicity and accuracy of MML and the speed of deep learning.

We note in passing that an earlier effort at combining MML with neural nets is [41]. We further note that some approaches to deep learning use a (suitably weighted) combination of a squared error term and a Kullback–Leibler divergence term. Given that squared error comes (or can come) from a Gaussian log-likelihood, this version of deep learning regularization bears similarities to D. F. Schmidt’s MML approximation [11] ([6] footnotes 64 and 65).(The MMLD version of MML ([6] Section 0.2.2, p. 528) [42] ([19] Sections 4.10, 4.12.2 and 8.8.2, p. 360) modified MML87 [9] to allow for cases when the Bayesian prior is not approximately constant over the relevant region. D. F. Schmidt’s MML approximation, just discussed, is a further modification, and explicitly introduces Kullback–Leibler divergence into the expression.) We also ask, for future work, whether our approach might be combined with graph neural networks [43] or (higher-dimensional) hyper-graph neural networks.

## 4. Long Short-Term Memory (LSTM)

With the development of computational power in electronic equipment, powerful computers provide many learning algorithms and approaches in time series forecasting [44,45,46]. Deep learning is one of the popular approaches in recent years; it provides a complex model that has at least the potential to capture (and often does capture) more general information from the predictors than a traditional model, such as ARMA. Long short-term memory (LSTM) is a special kind of recurrent neural network introduced by Hochreiter and Schmidhuber in 1997 [47]. LSTM manages the two state vectors, the short-term state $h_{t}$ and long term state $c_{t}$, and uses the gating mechanism by adding linear components from the previous layer in order to provide the long memory. LSTM has been widely used in time series forecasting because it is able to capture more information in the time series data, particularly for the financial econometrics area, where the price of financial assets depends on various different factors that are difficult to represent by a linear model [46,48]. Each LSTM layer, including the cells of the forget gate, input gate, and output gate, is shown in Figure 1.
Forget gate: $f_{t}=\sigma \left(U^{f}x_{t}+W^{f}h_{t-1}+b^{f}\right)$;Input gate: $i_{t}=\sigma \left(U^{i}x_{t}+W^{i}h_{t-1}+b^{i}\right)$;Output gate: $o_{t}=\sigma \left(U^{o}x_{t}+W^{o}h_{t-1}+b^{o}\right)$.

The forget gate uses a sigmoid function σ(x) from Equation (Equation 10). It has a value between 0 and 1, and it determines how much information should be forgotten. If the result from the sigmoid function is close to 0, then more information should be forgotten, and if the result from the sigmoid function is close to 1, then less information should be forgotten.
(10)$$\sigma \left(x\right)=\frac{1}{1+e^{-x}}$$

The input gate also uses the sigmoid function, the input gate controls the value input from the input function of $g_{t}=tanh\left(Wh_{t-1}+Ux_{t}+b\right)$ using the tanh(x) function:(11)$$tanh\left(x\right)=\frac{sinh(x)}{cosh(x)}=\frac{e^{x}-e^{-x}}{e^{x}+e^{-x}}$$

The input gate controls how much information should be remembered. The LSTM long-term state uses an element-wise operation with $c_{t}=f_{t}⊙c_{t-1}+g_{t}⊙i_{t}$, where ⊙ is element-wise multiplication (of two matrices of the same dimension), also known as the Hadamard product.

The output gate $o_{t}$ controls how much long-term information $c_{t}$ should be carried forward to the next layer, and it also contributes to the short-term state of $h_{t}$. The result from the output gate function is also between 0 and 1, and the LSTM short-term state also uses element-wise multiplication, with $h_{t}=o_{t}⊙tanh\left(c_{t}\right)$. An LSTM with more than one layer is shown in Figure 2, and its structure enables the LSTM to capture long-term and short-term information in order to forecast. As usual, an LSTM is trained by back propagation as other neural network models are. An LSTM requires time series data to train the model, and its time series pattern will be modeled in every layer of the network.

## 5. Hybrid ARMA-LSTM Model

In recent years, LSTM and its variants—along with some hybrid models—have been thought by many to largely dominate the financial time series forecasting domain [46]. The LSTM is able to capture the dependency of residuals across time, and the LSTM is trained by the time step [49]. In this paper, we are using the Moving Average lag order *q* from ARMA parameters selected by MML87, AIC, BIC, and HQ—if q=0, then we only use ARMA to forecast the time series data without LSTM. Our LSTM model is composed of a single input layer with an input shape of MA order and the sequence learning features. The following LSTM layer also contains the sequence learning features, and the third LSTM layer with the same unit is followed by the fourth dense layer with one unit.

We developed Algorithm 1 based on [17] using a different loss function and activation function in the regression task. The hybrid ARMA-LSTM model trains the LSTM model by the residuals from the ARMA model. (This is similar in spirit to the discussion in ([7] (Section 5.1))). The simple point here is that the LSTM has at least the potential to find dependencies that the ARMA model (on its own) can not express. In this paper, MML87, AIC, BIC, and HQ have been used to select the model parameter orders from the ARMA model; so, this paper not only compares the errors of the hybrid ARMA-LSTM model with those from the single ARMA model but also the hybrid model in terms of the selection(s) of MML87, AIC, BIC, and HQ. The forecast from the ARMA model is the fitted mean $\mu_{t+1}$. Because information is hidden in the residuals from the ARMA model (in a similar vein to ([7] (Section 5.1))), the forecast of the hybrid model will be
(12)$${\hat{Y}}_{t+1}=\mu_{t+1}+E_{t+1}$$
where $\mu_{t+1}$ represents the linearity modeling of data from the ARMA model selected according to the information-theoretic MML87, AIC, BIC, and HQ. The term $\epsilon_{t}$ is the residual left by the ARMA model $Y_{t}-{\hat{Y}}_{t}$, and $E_{t+1}=f\left(\epsilon_{t}\right)=f\left(Y_{t}-{\hat{Y}}_{t}\right)$, which is forecasted by the LSTM based on the past residual values $\epsilon_{t},\epsilon_{t-1},...,\epsilon_{t-q}$, where the parameter *q* is selected by MML87, AIC, BIC, and HQ. The hybrid ARMA-LSTM model combines both linear and non-linear tendencies in time series data [50].
**Algorithm 1** Algorithm 1 with the LSTM Model [17].**Require:** number of epochs = 10
 **while** MA(*q*) order in order set selected by MML, AIC, BIC, and HQ **do**  model.add(LSTM(30, return_sequences=True, input_shape=(*q*, 1)))  model.add(LSTM(30, return_sequences=True))  model.add(LSTM(30))  model.add(Dense(1))

The algorithm of the hybrid model is shown below (Algorithm 2):
**Algorithm 2** Algorithm 2 with the Hybrid ARMA-LSTM Model.**Require:** number of data *n* ≥ 0 **while**
*N* ≤ number of different simulations **do**  **while** *n* ≤ number of dataset in simulation **do**   **while** *i* ∈ MA orders selected from MML, AIC, BIC, and HQ **do**    **if** *i* ≠ 0 **then**      Train LSTM model by the residuals of ARMA model      Rolling forecast the residual by LSTM      Calculate root mean squared error by *Y*_*t*+1_    **else if** *i* = 0 **then**      Calculate root mean squared error by forecast from ARMA only

## 6. Experiments

The experiments have been designed to compare the results of the ARMA model itself with the hybrid ARMA-LSTM model and also to compare different versions of the hybrid model with the parameters variously selected by the MML87, AIC, BIC, and HQ. In order to analyze the accuracy of forecasting, we are using the root mean squared error, RMSE $=\sqrt{\frac{1}{T}\sum_{t=1}^{T}{\left(y_{t}-{\hat{y}}_{i}\right)}^{2}}$, to compare the different results, where *T* stands for the forecast window size, and we are using rolling forecast in this experiment. To elaborate and clarify, for the financial data in Section 6.2, we do integer differencing with d=1 to obtain stationarity before using the ARMA model and, as such, use an ARIMA or autoregressive integrated moving average model. We compare the performance of ARMA, ARMA-LSTM, and LSTM alone on simulated dataset(s) (Section 6.1) and also on real-world financial (Section 6.2) and air pollution (Section 6.3) datasets.

We argue elsewhere (([6] footnotes 75 and 76) ([25] Section 3) ([40] Section 4.1)) about various uniqueness and invariance properties of log-loss (or logarithm loss). Squared error is a popular method and is also a variant of log-loss.

### 6.1. Simulated Dataset(s)

In this section, we perform experiments using various previously described modeling methods on simulated data, and we begin (in terms of LNPPP space ([6] Section 0.2.7)) by describing the experiments. We use a uniform distribution on [−0.9,0.9] (from minimum −0.9 to maximum 0.9) to randomize the parameters *p* and *q* of ARMA(*p*,*q*) for the data simulation by using the *arima.sim* function in R and then reject them if they are outside the stationarity region. There are 5×2=10 different parameter sets from $p_{1},...,p_{5}$ and $q_{1},q_{2}$. The values in the table are the average RMSE over 100 runs (with standard deviation in brackets) in the simulated dataset corresponding to the particular parameters. The dataset includes *N* = 50, 100, 200, 300, and 500 time series data points in one dataset and also includes forecast windows of window size(s) T=3,10,30, and 50. Table 1 shows the average of RMSE trained by LSTM alone (with different numbers of LSTM time steps) with different forecast window sizes, *T*. The results suggest that the LSTM alone does not work well in ARMA simulated data. For convenience of reading, we have moved Table A1, Table A2, Table A3, Table A4, Table A5, Table A6, Table A7 and Table A8 to Appendix A; each value in Table A1 is the average RMSE of forecast errors over the datasets (with standard deviation in brackets). The bold texts indicate the smallest forecast errors from the different kinds of models. Table A1, Table A2, Table A3 and Table A4 provide a comparison of different forecast window sizes (or window size) with *T* = 3, 10, 30, and 50.

Table A2 shows the results for the average RMSE in the datasets for different simulated ARMA parameter sets, with the forecast window of T=10. Table A3 provides the comparison of root mean squared error results of those datasets in different criteria, also comparing different simulated datasets with the forecast window of T=30.

A large forecast window usually decreases the accuracy for the time series model. A window size of *T* = 50 (Data provided by Table A4) is 50% of the size of the in-sample set, and the MML87 hybrid model still outperforms its rivals. This indicates that the MML information criterion is efficient in model selection, and the algorithm of the hybrid model is also efficient in time series analysis, with the result of T=50, as shown in Table A4. Table 2 shows the average of the ten different parameters of the simulated dataset in the forecast window sizes of T=3 (Data provided by Table A1), 10 (Data provided by Table A2), 30 (Data provided by Table A3) and 50 (Data provided by Table A4) with the in-sample size of N=100.

MML87 outperforms the rival methods in the in-sample size of N=100 in all cases of T=3,10,30, and 50. MML87 not only considers the goodness of fit of data but also considers the model complexity. Figure 3 shows that MML87 has a lower root mean squared error in most cases. The hybrid model selected by MML87 has the lowest error rate for T=3,10, and 30. These comparisons argue well for MML. The results of N=100 with T=50 seem to suggest that for a large size of the forecast window, the complex hybrid ARMA-LSTM model seems to perform better than the simple time series model. Given that the simulated data were generated from an ARMA model, it is not immediately apparent why adding LSTM to produce a hybrid model should be advantageous in the case of larger datasets (although we would typically expect this if not dealing with data that are purely from an ARMA model). Table 3 shows the average of the ten different parameters of the simulated dataset in the in-sample size of *N* = 50 (Data provided by Table A5), 100 (Data provided by Table A2), 200 (Data provided by Table A6), 300 (Data provided by Table A7), and 500 (Data provided by Table A8).

Table A5, Table A6, Table A7 and Table A8 compare six different models or model selection techniques in the RMSE of the dataset in N=50,200,300, and 500, with the forecast window size T=10. AIC tends to overfit for small datasets, such as N=50 (Data provided by Table A5 in Appendix A). Through an increase in the amount for the in-sample dataset, the RMSE decreases in the hybrid ARMA-LSTM model because the larger size of data helps the LSTM to train and fit an accurate model. Thus, the results show the RMSE for the MML87 model is lower than the other models in the range N=100,200, and 300. Because of the efficiency in controlling the model complexity in MML87, the model can avoid the overfitting problem for small datasets.

The hybrid model with LSTM overfits when the in-sample size is small, basically because there is a larger amount of parameters that need to be estimated compared to the pure ARMA model. On the other hand, the hybrid model tends to perform well for a large in-sample size because the deep learning model is often better off for a large in-sample size, such as N=200 (Data provided by Table A6), 300 (Data provided by Table A7), and 500 (Data provided by Table A8).

For a small in-sample size, such as N=50, the BIC performance is good on the hybrid ARMA-LSTM because BIC is able to select the model well without overfitting. The MML87-Hybrid has the smallest average RMSE for N=100,200, and 300 for the different randomized datasets. The hybrid models work efficiently when there is enough in-sample data; otherwise, it can also overfit small datasets. In the meantime, by comparing the RMSE from MML87-ARMA, AIC-ARMA, BIC-ARMA, and HQ-ARMA, the results favor MML87 rather than AIC, BIC, and HQ. MML87 has a good performance in time series model selection and is able to select the ARMA model with lower forecasting errors. However, as noted earlier in this section, given that the simulated data were generated from an ARMA model, it is not immediately apparent why adding LSTM to produce a hybrid model should be advantageous in the case of larger datasets (although we would typically expect this if not dealing with data that are purely from an ARMA model). Figure 4 shows the comparison of RMSE in the in-sample size *N* = 50, 200, 300, and 500.

### 6.2. Financial Data-and Extension to ARIMA Models

Stock return prediction is one of the most popular research topics in economics and finance [51,52]. This section studies the performance of the hybrid model from MML87; the hybrid models from AIC, BIC, HQ; and the ARIMA models selected by MML87, AIC, BIC, and HQ. The stock prices were selected from the components of the Dow Jones Industrial Average, including Apple (APPL), Boeing (BA), Cisco System (CSCO), Goldman Sachs (GS), IBM, Intel (INTC), Johnson & Johnson (JNJ), JPMorgan Chase (JPM), Coca-Cola (KO), and 3 M (MMM).The data selected start at 23 September 2016 and finish at 22 September 2021, with a total of 1258 trading days. This experiment studies the different performances in forecast window sizes T=3,5,10,30,50,70,100,130,150, and 200. Table 4 shows the characteristic of stock prices selected, including mean, standard deviation, and partial autocorrelation.

The empirical results show that the hybrid ARIMA-LSTM model can substantially outperform the traditional ARIMA (Autoregressive Integrated Moving Average) time series model, particularly in the forecast window sizes of T=5,30,100,130,150, and 200. Many studies demonstrated that the stock return depends on various factors, such as dividend yield, the book to market ratio, and/or interest rate [51,53,54]. However, traditional linear time series models are not able to take into account the effect of all those factors, thus requiring a more complex model to capture the information hidden in residuals from the ARIMA model. The hybrid model with LSTM is able to model publicly available and other information, which we have no reason to believe will be restricted, coming from a purely ARMA or ARIMA model. In order to make the stock price stationary in time series analysis, the ARIMA models are using the parameter d=1 (or, equivalently, first-order differencing). As the experimental results show, MML87 outperforms the other information-theoretic criteria AIC, BIC, and HQ in terms of lower root mean squared error for out-of-sample forecasting. Figure 5 demonstrates the log prices for stock prices selected in this experiment.

The hybrid model tends to outperform for a large forecast window size rather than the small forecast window size because a large lookahead in forecasting has higher uncertainty. For much—or perhaps even most—of the financial industry, there is high volatility in long forecasts. The notion of semi-strong market efficiency suggests that the stock price fully and fairly reflects publicly available information in the time horizontal in the forecast window and also reflects all past information (although by no means all authors agree with this in its entirety [55], partly due to principles of Solomonoff [39] and Wallace [7]). Thus, it is more likely that a complex model will at least be able to provide accurate results in predictions for a T greater than 100. Table 5 shows MML models have lower the RMSE in most cases for different forecast window sizes in financial data.

Table 6 provides the average of RMSE for the selected stocks in different sizes, *T*, of the forecast window (shown in different columns) and numbers of LSTM time steps (shown in different rows). The LSTM models are trained by scalers in the range of 0 to 1, and the LSTM model performs worse in the case without scaling, which indicates that the neural network LSTM is scale insensitive and that combining the traditional ARMA time series model makes the neural network more scale-sensitive [56]. The results from Table 6 suggest that the LSTM model alone (unenhanced by ARMA and ARIMA) is not particularly able to capture the time series pattern for the stock price. The figures of the average RMSE are significantly higher than traditional ARMA and ARMA-LSTM models. Figure 6 shows the comparison between the ARIMA model and the hybrid ARIMA-LSTM model in this experiment.

### 6.3. PM2.5 Pollution Data

In this section, we use environmental data of PM2.5 pollution levels in the city of Beijing, China, with ten sensors located in different areas. The data are hourly PM2.5 levels in 53 days in 2013.

We are using the same data length and information-theoretic methods from Section 6.2 in order to demonstrate the performance of MML compared to rival methods. Table 7 shows the comparison between MML, AIC, BIC, and HQ. The hourly PM2.5 data have a seasonality; the level of PM2.5 reaches its highest near midday and decreases to its lowest near midnight. The results suggest that MML is a good model selection technique in this case.

Table 8 shows the LSTM model alone in the PM2.5 data, and the results suggest that the LSTM model (on its own, unenhanced by ARMA and ARIMA) outperforms in the smaller-sized forecast windows, such as T=3,5, and 10. The RMSEs in larger window sizes (T≥50) are much larger for the LSTM than for the ARMA model and hybrid ARMA-LSTM.

## 7. Conclusions

We have investigated time series modeling in the Minimum Message Length framework using Wallace and Freeman’s (1987) approximation [9]. The hybrid ARMA-LSTM model has been compared with the traditional ARMA (Autoregressive Moving Average) time series model based on the information-theoretic approaches: AIC, BIC, HQ and MML87. We performed experiments on simulated data and also on two real-world datasets (financial and environmental data). We conducted the experiments based on hybrid ARMA-LSTM (with LSTM) and ARMA without LSTM (long short-term memory). This could be broadly thought of as constituting two experiments each on three datasets or with six experiments. For each of the six experiments, the results show that MML87 outperforms the other information-theoretic criteria. The hybrid ARMA-LSTM model performs better than the traditional ARMA model, and the MML hybrid ARMA-LSTM model performed best out of everything considered. It is worth noting that the LSTM model alone with unscaled data performed worse than everything else considered. In summary, MML87 is able to select the lower forecasting errors better than the AIC, BIC, and HQ, as the experimental results show.

## Figures and Tables

**Figure 1 entropy-23-01601-f001:**
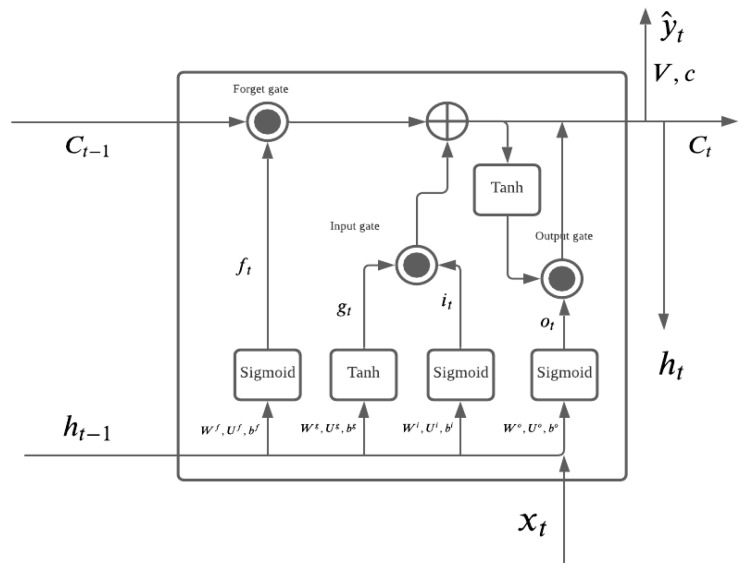
LSTM Structure.

**Figure 2 entropy-23-01601-f002:**
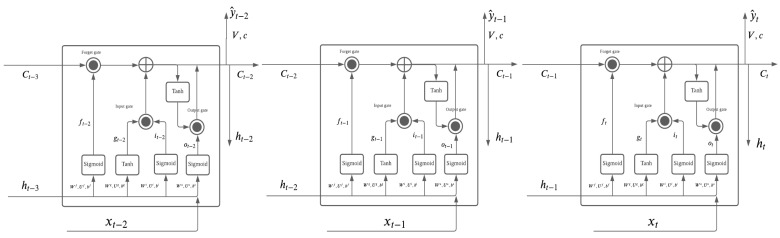
LSTM Overlapping.

**Figure 3 entropy-23-01601-f003:**
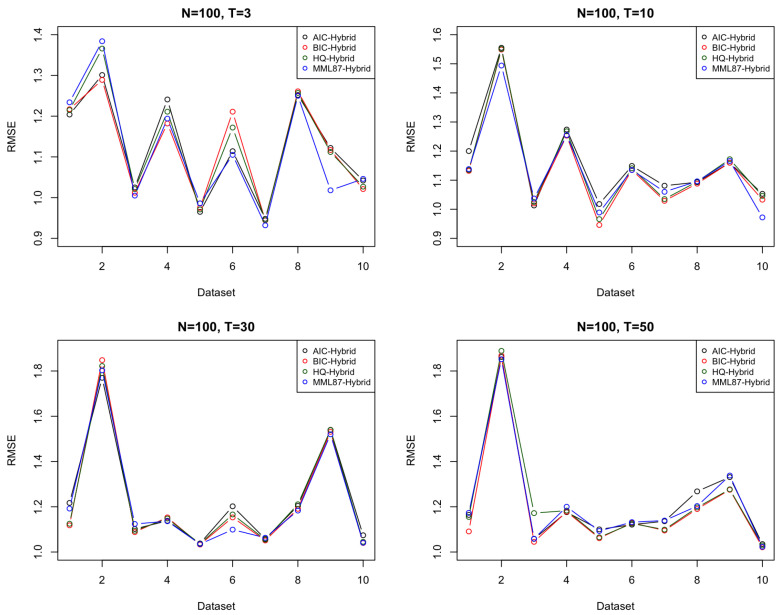
Comparison in forecast window sizes *T* = 3, 10, 30, and 50.

**Figure 4 entropy-23-01601-f004:**
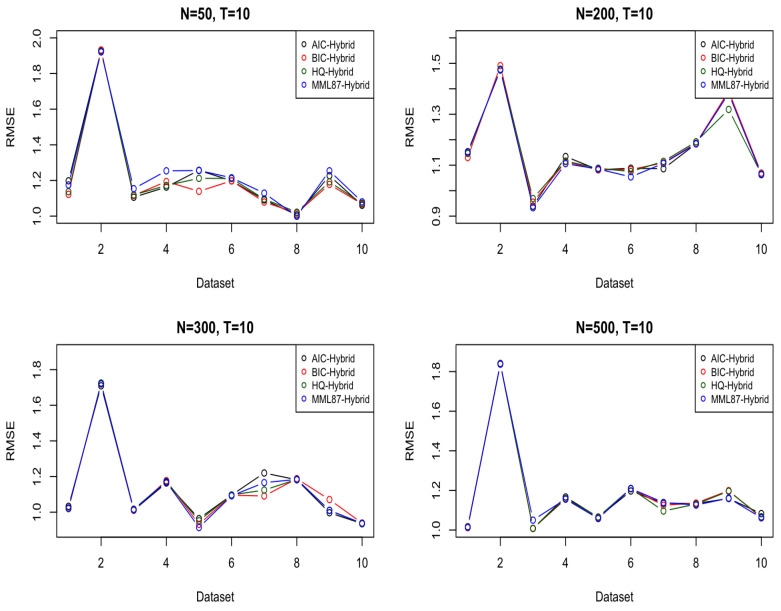
Comparison in the in-sample size *N* = 50, 200, 300, and 500.

**Figure 5 entropy-23-01601-f005:**
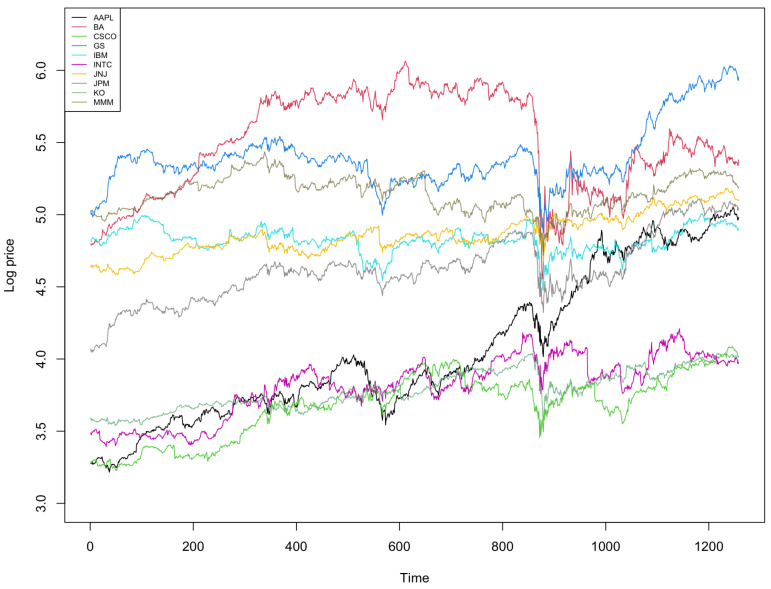
Log prices for ten selected stocks.

**Figure 6 entropy-23-01601-f006:**
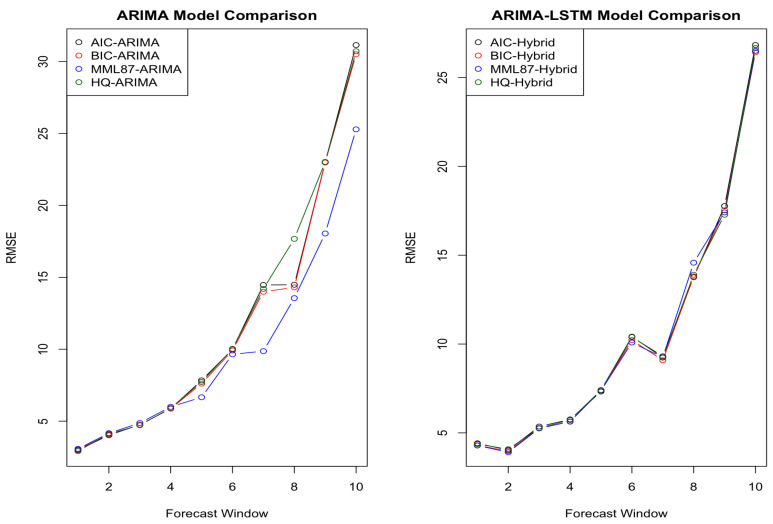
Comparison in different forecast windows.

**Table 1 entropy-23-01601-t001:** RMSE in LSTM for simulated data ($p_{1},q_{1}$) with different time steps and *N* = 100.

No. of LSTM Time Steps	T=3	T=10	T=30	T=50
1	1.2519	1.3677	1.4962	1.3911
2	1.1794	1.2442	1.3863	1.2718
3	1.3372	1.6324	1.2256	1.3018
4	1.2195	1.2301	1.3284	1.3951
5	1.1341	1.6294	1.4276	1.4494

**Table 2 entropy-23-01601-t002:** Average of RMSE in forecast window size *T* = 3, 10, 30, and 50.

Average of RMSE								
	ARMA				ARMA-LSTM			
	AIC	BIC	HQ	MML87	AIC	BIC	HQ	MML87
*T* = 3	1.086	1.076	1.090	**1.072**	1.121	1.123	1.134	**1.115**
*T* = 10	1.149	1.136	1.144	**1.121**	1.159	1.136	1.143	**1.134**
*T* = 30	1.338	1.331	1.340	**1.325**	1.234	1.221	1.224	**1.220**
*T* = 50	1.308	1.296	1.297	**1.295**	1.225	**1.195**	1.219	1.221

**Table 3 entropy-23-01601-t003:** Average of RMSE for in sample size *N* = 50, 100, 200, 300, and 500 with forecast window size *T* = 10.

Average of RMSE (and Standard Deviation)								
	ARMA				ARMA-LSTM			
	AIC	BIC	HQ	MML87	AIC	BIC	HQ	MML87
*N* = 50	1.301	1.291	1.299	**1.280**	1.224	**1.202**	1.216	1.244
*N* = 100	1.149	1.136	1.144	**1.121**	1.159	1.136	1.143	**1.134**
*N* = 200	1.177	1.187	1.189	**1.183**	1.159	1.161	1.164	**1.154**
*N* = 300	1.163	1.152	1.155	**1.147**	1.131	1.125	1.124	**1.123**
*N* = 500	**1.194**	1.197	1.1984	1.196	1.180	**1.173**	1.179	1.181

**Table 4 entropy-23-01601-t004:** Mean, standard deviation, PACF lag 1 to 3 for ten selected stocks.

	Mean	S.D	PACF1	PACF2	PACF3
AAPL	66.440217	37.060808	0.996875	0.044454	−0.004848
BA	258.704781	82.478194	0.995870	−0.031231	−0.061804
CSCO	40.585947	8.595774	0.994585	0.073202	−0.016488
GS	227.095242	56.820929	0.993579	0.039741	−0.043412
IBM	124.851224	10.369478	0.982339	0.070195	−0.040622
INTC	46.269478	9.305502	0.992194	0.178757	−0.053398
JNJ	130.715314	18.399352	0.993930	0.050988	−0.031304
JPM	104.046116	24.467471	0.993854	0.067756	−0.049235
KO	44.519034	6.089778	0.993828	0.031639	−0.039178
MMM	173.550240	20.467854	0.991641	0.004475	0.026664

**Table 5 entropy-23-01601-t005:** RMSE for forecast window sizes *T* = 3, 5, 10, 30, 50, 70, 100, 130, 150, and 200.

Average of RMSE (& Standard Deviation)								
	ARIMA				ARIMA-LSTM			
	AIC	BIC	HQ	MML87	AIC	BIC	HQ	MML87
*T* = 3	**2.987** (3.446)	3.027 (3.555)	2.914 (3.567)	3.075 (3.572)	4.414 (4.75)	4.302 (4.608)	4.375 (4.757)	**4.289** (4.616)
*T* = 5	**4.024** (5.091)	4.077 (5.228)	4.126 (5.218)	4.163 (5.086)	4.024 (5.45)	3.966 (5.42)	4.081 (5.739)	**3.907** (5.449)
*T* = 10	4.748 (4.707)	**4.747** (4.858)	4.712 (4.815)	4.868 (4.347)	5.359 (5.429)	5.261 (5.268)	5.272 (5.443)	**5.249** (5.262)
*T* = 30	5.872 (6.797)	**5.867** (6.6)	5.91 (6.576)	5.994 (5.662)	5.754 (4.822)	**5.628** (4.687)	5.726 (4.776)	5.643 (4.677)
*T* = 50	7.834 (7.511)	7.609 (7.298)	7.726 (7.269)	**6.659** (6.966)	**7.328** (6.787)	7.411 (6.879)	7.405 (6.789)	7.384 (6.898)
*T* = 70	9.991 (9.491)	9.909 (9.316)	10.024 (9.173)	**9.645** (7.99)	10.393 (8.048)	10.221 (7.789)	10.42 (8.061)	**10.085** (7.612)
*T* = 100	14.465 (17.187)	13.991 (15.428)	14.197 (13.637)	**9.866** (10.854)	9.304 (9.256)	**9.087** (9.35)	9.235 (9.396)	9.253 (9.486)
*T* = 130	14.482 (9.714)	14.301 (10.571)	17.672 (13.139)	**13.551** (10.238)	**13.768** (10.598)	13.811 (11.124)	13.9 (11.516)	14.581 (10.972)
*T* = 150	22.985 (28.173)	22.985 (28.077)	23.021 (28.071)	**18.045** (17.856)	17.778 (16.771)	17.526 (16.582)	17.98 (16.734)	**17.461** (15.931)
*T* = 200	31.144 (37.567)	30.502 (38.314)	30.712 (38.322)	**30.286** (32.564)	26.831 (31.63)	**26.424** (31.547)	26.662 (31.645)	26.507 (31.59)

**Table 6 entropy-23-01601-t006:** LSTM with different time steps for financial data in varying forecast windows.

No. Steps	*T* = 3	*T* = 5	*T* = 30	*T* = 10	*T* = 50	*T* = 70	*T* = 100	*T* = 130	*T* = 150	*T* = 200
1	8.5789	10.1965	56.7817	104.3681	123.4805	119.2673	151.1338	107.2951	114.8106	73.2335
3	5.7604	3.5166	3.6097	13.325	10.6368	31.9361	33.4419	26.0112	31.5578	26.6354
5	4.0695	3.0575	8.5064	11.9009	15.5075	17.3077	19.0942	48.0012	30.0622	36.6099
7	3.9708	6.4145	10.6368	6.8547	13.2163	16.5474	19.0724	32.7076	20.5954	44.1875
10	5.3985	6.4576	5.9597	13.8295	16.0972	20.6271	12.8859	28.2251	28.2803	25.5409

**Table 7 entropy-23-01601-t007:** RMSE for forecast window sizes *T* = 3, 5, 10, 30, 50, 70, 100, 130, 150, and 200.

Average of RMSE & Standard Deviation								
	ARIMA				ARIMA-LSTM			
	AIC	BIC	HQ	MML87	AIC	BIC	HQ	MML87
*T* = 3	26.805 (7.496)	26.689 (7.532)	26.569 (7.545)	**23.104** (7.843)	25.768 (7.833)	23.066 (8.693)	24.791 (7.883)	**22.965** (6.711)
*T* = 5	28.036 (6.986)	27.538 (6.805)	27.479 (7.186)	**23.478** (8.426)	24.636 (8.518)	22.309 (7.596)	24.113 (8.584)	**21.666** (7.424)
*T* = 10	30.633 (12.679)	31.502 (12.283)	31.585 (12.518)	**30.074** (14.917)	26.970 (10.502)	27.566 (13.487)	27.924 (10.107)	**25.102** (9.855)
*T* = 30	40.730 (14.001)	40.788 (13.372)	40.157 (14.195)	**37.989** (19.180)	31.022 (11.409)	29.382 (12.196)	31.689 (14.124)	**28.572** (12.229)
*T* = 50	39.097 (4.238)	**38.662** (4.660)	39.007 (4.232)	42.986 (6.062)	35.639 (5.599)	**33.335** (5.184)	36.036 (6.339)	40.568 (9.24)
*T* = 70	34.004 (4.223)	33.551 (4.105)	34.773 (3.514)	**32.030** (9.404)	48.942 (12.377)	45.305 (10.567)	49.723 (12.068)	**42.987** (8.705)
*T* = 100	32.002 (2.434)	32.444 (2.425)	**31.170** (2.865)	37.925 (4.444)	56.024 (13.199)	51.543 (12.714)	59.705 (13.435)	**49.513** (11.45)
*T* = 130	44.023 (2.583)	44.162 (2.576)	43.635 (2.836)	**43.802** (1.853)	36.183 (7.184)	**33.716** (4.591)	39.401 (8.488)	46.496 (7.168)
*T* = 150	44.463 (1.612)	44.736 (1.862)	43.928 (1.773)	**41.150** (5.221)	32.574 (6.211)	31.225 (4.301)	**30.923** (6.598)	33.584 (7.679)
*T* = 200	42.150 (2.620)	42.372 (2.522)	**41.863** (2.787)	43.75 (4.07)	46.711 (13.86)	**43.721** (11.349)	53.393 (12.458)	46.363 (12.472)

**Table 8 entropy-23-01601-t008:** LSTM for PM2.5 Beijing data in different time steps and forecast windows.

No. Steps	*T* = 3	*T* = 5	*T* = 10	*T* = 30	*T* = 50	*T* = 70	*T* = 100	*T* = 130	*T* = 150	*T* = 200
1	3.0976	5.7806	16.5048	47.8431	53.7436	67.2412	81.7044	92.6897	73.4192	71.5536
3	4.4983	8.1565	17.0462	34.0492	36.3896	47.2558	64.68533	90.7986	78.9972	78.5648
5	4.7719	9.2208	18.7955	33.6065	50.4786	56.7465	59.3666	75.4321	102.0098	88.1695
7	5.9126	9.8355	15.3696	25.8551	38.6874	53.06845	50.962	87.998	92.101	101.2337
10	8.4289	11.4749	11.4479	38.3303	44.5138	65.6299	70.6415	74.6879	90.1211	84.0196

## Data Availability

Not applicable.

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
