# Peer review of "Minimum Message Length in Hybrid ARMA and LSTM Model Forecasting"

_entropy, 2021, doi:10.3390/e23121601_

Round 1

Reviewer 1 Report

In this paper, the authors have merged ARMA and LSTM models to come up with ARMA-LSTM model. The paper is well written and easy to understand. I have not identified many typos.  

However, I request the authors to make necessary changes/improvements in the draft.

  1. Please highlight the novelty of your work followed by how challenging is the task. It seems that you are simply passing the data generated by ARMA into LSTM for learning ad calling it as ARMA-LSTM model.

          Please explain why it is non-trivial or challenging to push ARMA into LSTM.

  2. The experimental results are very very weak. 

         2. 1. Please evaluate your model not only against ARMA, but also LSTM also.

         2.2. Conduct more experiments using many datasets, especially employing real-world datasets.

         2.3. There are many recent survey papers on machine learning + time series. Please read them, and refer them in their work. Compared your algorithm with the existing ML/DL techniques, not just with age-old ARMA from stats.

Reviewer 2 Report

I think that the paper requires extensive re-reading and corrections before it can be published. Just some examples of corrections that are needed:

  1. Lines 20-21: ''The ARIMA model ... the stationery." Please read again and correct the whole sentence!
  2. 35: ''from the publication ...[5]'' doesn't sound good...
  3. 86: ''let fi'': ?
  4. 102: what is \pi? ('' there is prior knowledge of \pi'')
  5. 123: ''has been widely use in the time series forecasting''-->''has been widely used in time series forecasting''. In the same sentence there are other grammar mistakes...
  6. 130: ''from equation 11''?
  7. Algorithm 1 is cited in the paper?

and so on...

Reviewer 3 Report

Dear authors,

My major concern is that you do not compare your proposed method against other state-of-the-art. I would ask to make some experiments reproducing other methods found in the  literature (under the same financial dataset) and compare their work against yours.

There are also some minor comments that needs to be addressed prior publication. You can find them in the attached document  

Round 2

Reviewer 1 Report

The authors have addressed all the comments raised in my previous review. 

Reviewer 2 Report

Please read the attached file

Reviewer 3 Report

The changes made by authors resulted to a significant improvement of the manuscript. Please correct the following:

Line 8: Bayesian information8 -->Bayesian information
Lines 44-45: deep learning also including serval variants --> deep learning also includes several variants
Line 48: for 52 data --> for data
Line 55: outperfors compare to the AIC… --> outperforms when compared to AIC…

Also please check please proofread the whole manuscript and use past tense
